# Characteristic of Stimulus Frequency Otoacoustic Emissions: Detection Rate, Musical Training Influence, and Gain Function

**DOI:** 10.3390/brainsci9100255

**Published:** 2019-09-26

**Authors:** Yao Wang, Zhihang Qi, Mengmeng Yu, Jinhai Wang, Ruijuan Chen

**Affiliations:** Department of Biomedical Engineering, School of Life Sciences, Tianjin Polytechnic University, Tianjin 300387, China; wangyao_show@163.com (Y.W.); qizhihang0630@163.com (Z.Q.); MistyYu163@163.com (M.Y.); wangjinhai@tjpu.edu.cn (J.W.)

**Keywords:** auditory plasticity, detection rate, gain function, musical training, stimulus frequency otoacoustic emissions

## Abstract

Stimulus frequency otoacoustic emission (SFOAE) is an active acoustic signal emitted by the inner ear providing salient information about cochlear function and dysfunction. To provide a basis for laboratory investigation and clinical use, we investigated the characteristics of SFOAEs, including detection rate, musical training influence, and gain function. Sixty-five normal hearing subjects (15 musicians and 50 non-musicians, aged 16–45 years) were tested and analyzed at the probe level of 30 and 50 dB sound pressure levels (SPL) in the center frequency of 1 and 4 kHz in the study. The results indicate that (1) the detection rates of SFOAE are sensitive to the gender, (2) musicians reveal enhanced hearing capacity and SFOAE amplitudes compared with non-musicians, and (3) probe frequency has a significant effect on the compression threshold of SFOAE. Our findings highlight the importance of SFOAE in the clinical hearing screening and diagnosis and emphasize the use of musical training for the rehabilitation enhancement of the auditory periphery and hearing threshold.

## 1. Introduction

Since the discovery of otoacoustic emission (OAE) in 1978 [1], several studies have attempted to demonstrate OAE as a promising non-invasive tool for assessment of cochlear functional status [2,3]. However, the applicability of stimulus-frequency (SF) OAEs in laboratory investigation and clinical use (e.g., the detection rate and gain function under different conditions) is still unclear. Bidelman et al. showed an experience-dependent enhancement in the peripheral frequency resolution for musicians [4]. It needs to be further investigated whether the peripheral threshold (indicated by SFOAE amplitude) improved musical training. To explore the above questions, we investigated detection rate, musical training influence, and gain function of SFOAEs.

OAE is a kind of weak acoustic signal non-invasively recorded in the external auditory canal by a tiny microphone. It is produced from the cochlea and transmitted via the ossicles chain and the tympanic membrane [5]. OAE is a by-product of the active mechanism of the cochlea and can reflect the function of outer hair cells (OHCs) [1,6]. Therefore, it is sensitive to the damage of the inner ear. OAEs are divided into spontaneous otoacoustic emission (SOAE) and evoked otoacoustic emission (EOAE) according to whether the emission is induced by a stimulus or not. SOAEs are recordable in the absence of deliberate external acoustic stimulation [7,8]. EOAEs are measured in the ear canal in the case of external stimulation. Owing to the different types of stimuli, EOAE includes distortion product otoacoustic emission (DPOAE), transient evoked otoacoustic emission (TEOAE), and SFOAE. SFOAE is an active acoustic signal emitted by the inner ear with the same frequency as the stimulus, and is generally considered the result of activities of the cochlear mechano-electrical transducer [9,10]. It can provide salient information about cochlear function and dysfunction objectively and non-invasively [11,12,13]. Since the frequency of SFOAE is identical to the stimulus in terms of time and frequency domains, SFOAEs are expected to show peripheral auditory function at specific sites of cochlea. 

Many investigations have revealed that OAE is a sensitive acoustic signal, and it can indicate cochlear function in subjects with normal hearing or moderate hearing impairment [14,15]. To date, the detection rates of TEOAE, DPOAE, and SOAE have been primarily investigated. The detection rate of SOAEs only reaching 50%–74% mainly occurs at a specific frequency, and there are great individual differences in SOAEs [16]. Age and gender affect the DPOAEs prominently in normal hearing subjects. The detection of DPOAEs and mean amplitudes decreases with advancing age [17]. In terms of TEOAE, the study of detection rate is focused on newborns and tinnitus. The detection rate of TEOAEs is nearly 100% in normal-hearing participants at the age <60 [18]. However, the detection rate of TEOAE decreases with increasing age in normal hearing [pure tone audiometry <25 dB hearing level (HL) in 250~8000 Hz], suggesting that TEOAE is more sensitive than pure tone audiometry (PTA) in detecting early cochlear impairment [19]. In newborns, the general prevalence of TEOAE is 89%, with the average prevalence being 74% in newborns 0–3 day old and 97% in newborns 4–8 day old. Moreover, the prevalence of females is higher than males [20]. In tinnitus, the detection rates of TEOAE, DPOAE, and SOAE are all lower than that of the normal hearing group, also revealing that OAEs are sensitive to the early impairment of cochlea [21]. Previous studies have revealed that OAEs are sensitive in normal hearing subjects and abnormal groups [21,22]. It suggests that OAEs can be used for monitoring the progression of cochlear damage during the early stages of hearing impairment. Although SFOAE has a good frequency specificity compared with SOAE, TEOAE, and DPOAE, investigations of the detection rate of SFOAE have been rare to date.

As musical training is more complex and multimodal than most other daily life activities, music has increasingly been used as a tool for the investigation of human cognition and its underlying brain mechanisms [23,24,25]. Musicians exhibit enhanced sensory and cognitive abilities that impart notable advantages compared with non-musicians for pitch discrimination [26,27,28], speech in noise tasks [29,30,31], and auditory attention [32]. Several studies have suggested that such enhanced auditory skills in musicians are ascribed to long-term musical training, which induces functional [33] and structural [34] differences of the central auditory system. In other words, musical training influences the brain plasticity of musicians [35]. As early as 1970, Soderquist et al. reported that non-musicians are inferior to professional musicians in their ability to analyze complex waveforms [36]. In 2011, Bidelman et al. pointed out that musicians have significant tonal resolution advantages compared with non-musicians in both electrophysiological and behavioral tests [37]. This indicates that the enhancement effect of auditory brainstem neurophysiological response, which is induced by long-term musical training starting from childhood, has shifted to the cognitive level [38,39,40]. Although many studies indicate that musical training is related to auditory perception, it remains unclear whether it influences the intensity sensitivity of the auditory periphery. Many studies manifest that music training can enhance medial olivocochlear system (MOC) activity [41,42]. Owing to the nerve axons of the MOC being connected with the outer hair cells of the cochlea, internal changes of the cochlea of musicians may ensue [43]. Consequently, we hypothesize that long-term musical training may enhance the SFOAE amplitude.

A further question that we intended to explore was the gain function of SFOAE under different conditions to explore the possibility of clinical application of SFOAEs. The gain function is considered a measure of emission strength, which is plotted as OAE level in dB as a function of stimulus level [in dB sound pressure levels (SPL)] [44]. It reveals the maximum OAE gain and compression nonlinearity and can reflect the cochlear-mechanical response growth invasively [45]. The nonlinearity is attributed to outer hair cell (OHC) function, and it is present in normal hearing ears. Presumably due to OHC loss, the nonlinearity is decreased or absent in ears with hearing loss [45,46]. Thus, the gain function is of great importance in describing auditory deficits [47].

In this study, we investigated the characteristics of SFOAEs, including the detection rate, the musical training influence, and the gain function, in order to provide a basis for laboratory investigation and clinical use. By detecting and analyzing the PTA and the SFOAE amplitude in 65 normal hearing subjects (15 musicians and 50 non-musicians) at the probe levels of 30 and 50 dB SPL in the center frequency of 1 and 4 kHz, we explored and predicted (1) the detection rate of SFOAE under the conditions of different genders, probe levels, and probe frequencies; (2) that musical training may enhance the hearing threshold and SFOAE amplitude; and (3) that it is possible that the gain function of SFOAE is sensitive to the stimulus frequency.

## 2. Materials and Methods

### 2.1. Participants

This study consisted of three experiments in which all participants were native speakers of Chinese, right-handed according to the standard handedness questionnaire [48], and with normal hearing (≤20 dB hearing level for 250–8000 Hz); none had a history of hearing disorders or strong spontaneous optoacoustic emissions. During the experiments, the tested ear was randomly determined. All subjects gave their informed consent for inclusion before they participated in the study. The study was conducted in accordance with the Declaration of Helsinki, and the protocol was approved by the Shandong University institutional review board.

• Experiment 1: Detection Rates

Fifty young adults (26 females and 24 males) aged 16–45 years (mean ± standard deviation 23.94 ± 4.57 years) with normal hearing participated in the first experiment, and no musician was included in the 50 subjects.

• Experiment 2: Musical Training Influence

Thirty young adults including 15 musicians (12 females and 3 males) and 15 non-musicians (10 females and 5 males) aged 21–44 years (mean ± standard deviation 25.20 ± 4.32 years) participated in the second experiment. Based on self-reporting, all musicians commenced their musical training before the age of 10 (9.60 ± 2.77 years) and had consistently played a musical instrument or had vocal music training throughout their lives for 15 years (14.07 ± 2.84 years). The definition of “musician” was consistent with the previous study [49]. Non-musicians had <2 years of formal musical training throughout their lifespan. Musicians and non-musicians were matched in age (musicians, 25.93 ± 5.57 years; non-musicians, 24.47 ± 2.53 years) and formal education [49,50] (musicians, 17.93 ± 1.44 years; non-musicians, 18.13 ± 2.26 years) (for detailed information about musicians and non-musicians, see Appendix A).

• Experiment 3: Gain Function

Ten normal-hearing adults (6 females and 4 males) aged 20–27 years (mean ± standard deviation 22.50 ± 2.72 years) participated in the third experiment.

All the normal hearing participants without musical training in Experiments 2 and 3 were also included in Experiment 1.

### 2.2. Instruments

All experiments were carried out with participants sitting on a chair quietly in a sound-attenuating booth. Briefly, stimuli were generated by an external soundcard (Fire face 802, RME, Haimhausen, Germany) and delivered by a pair of miniature earphones (ER-3C, Etymotic Research, Elk Grove Village, IL, USA). One miniature earphone produced a constant tone as a probe, and the other miniature earphone produced a tone-burst as a suppressor. The acoustic signals in the ear canal were collected by a miniature microphone (ER-10B+, Etymotic Research) and amplified by 20 dB (ER-10B+ preamplifier, Etymotic Research). A monaural earplug including both miniature earphones and a miniature microphone was inserted into the ear canal of each participant. The measurement system was calibrated with a Total Sound ear simulator (type AE400). During the detection of SFOAE, the participants were able to watch a silent film. In the detection of PTA, each participant was instructed to pay attention to the probe tone by pressing a button.

### 2.3. Design and Procedure

Before the detection of SFOAE, PTA and SOAE tests were examined to select eligible subjects. It was guaranteed that there were no SOAEs in the frequency range of interest in order to avoid interference with the SFOAE. In Experiment 1, the detection rate was calculated as the number of participants who detected SFOAE divided by the total number of participants. The criteria for the presence of SFOAE was signal-to-noise-ratio ≥6 dB in the SFOAE spectra (an example of the presence of SFOAE is shown in Figure 1). The SFOAE amplitudes were tested under 4 conditions: 2 probe levels × 2 probe frequencies. The probe levels (*L*_p_) were 30 and 50 dB SPL, and center frequencies were 1 and 4 kHz. In Experiment 2, SFOAE fine structure was a high-resolution (40 Hz steps) SFOAE amplitude recording which was evoked in the probe frequency (*f*_p_) range of ± 200 Hz (relative to the center frequency, CF). Each subject was tested at the probe level (*L*_p_) of 30 dB SPL and CFs of 1 and 4 kHz. In Experiment 3, SFOAE amplitudes were plotted as a function of *L*_p_ (5–50 dB SPL in 5 dB steps) in SFOAE input/output function testing to determine the gain function of SFOAE. The gain function is considered a measure of emission strength and is plotted as OAE level in dB regarding stimulus level (in dB SPL).

### 2.4. Data Analysis

All data first underwent Kolmogorov–Smirnov testing to verify the compliance of normal distribution. All data analyses were conducted in SPSS 19.0 software (SPSS, Inc., Chicago, IL, USA). *p* < 0.05 was considered statistically significant.

#### 2.4.1. Experiment 1: Detection Rates

The detection rates of SFOAE were analyzed with chi-square tests among the factors of gender (female versus male), probe frequency (1 versus 4 kHz), and probe level (30 versus 50 dB SPL). The detection rates of SFOAE between groups of female and male were compared under 4 conditions: 2 probe levels × 2 probe frequencies. In addition to the *p* value, Cohen’s *d* was used to describe the effect size (ES) when the data satisfied the normal distribution.

#### 2.4.2. Experiment 2: Musical Training Influence

Mean values were provided first to compare the values of PTA and SFOAE amplitudes. Then, to ascertain whether musical training affects the inner ear, PTA at octave frequencies from 250 to 8000 Hz and SFOAE amplitudes at the center frequency of 4 kHz were analyzed with non-parametric statistics (two-factor Scheirer–Ray–Hare test [51]) as the values of them did not satisfy the normal distribution. Additionally, SFOAE amplitudes at the center frequency of 1 kHz were analyzed with two-factor analysis of variance (ANOVA) tests as the values of them satisfied the normal distribution. One factor was the group difference between musicians and non-musicians, and the other factor was probe frequency (250/500/1000/2000/4000/8000 Hz). Finally, simple effect analysis was conducted when significant interactions existed between the two factors of group difference and probe frequencies. In addition to the *p* value, partial eta squared (*ƞ*_p_^2^) was used to describe the effect size.

#### 2.4.3. Experiment 3: Gain Function

Three SFOAE metrics could be derived from the fitted curve of gain function: (1) peak strength (PS), the SFOAE level of the flat portion of the gain function; (2) compression threshold (CT), the stimulus level at which SFOAE strength began to decrease compressively; and (3) compression slope (CS), which indicated the rate of compression of the SFOAE gain function at probe levels beyond the CT [44]. The above three SFOAE metrics (see illustration in FIG3B of Ref [44]) were analyzed with independent-sample *t* tests between probe frequencies of 1 and 4 kHz and non-parametric statistics (Mann–Whitney U tests) when the data did not satisfy the normal distribution. In addition to the *p* value, Cohen’s *d* was used to describe the ES when the data satisfied the normal distribution.

## 3. Results

### 3.1. Detection Rate

Table 1 represents the detection rates of SFOAE that were tested with the different factors of gender, probe frequency, and *L*_p_ value. Figure 2a–d illustrates the detection rates of SFOAE between groups of female and male under four conditions: 2 probe levels × 2 probe frequencies. As shown in Table 1 and Figure 2a, among a total of 50 participants (24 females and 26 males), SFOAEs were detected in 17 female participants (71%) and 24 male participants (92%). The male detection rate of SFOAE was obviously higher than that of the female detection rate at the *f*_p_ of 1 kHz and *L*_p_ of 30 dB SPL, and a significant difference existed between them (χ^2^ = 3.899, *p* = 0.048, ES = 0.583). As shown in Table 1 and Figure 2b, SFOAEs were detected in 23 female participants (96%) and 25 male participants (96%). The statistical results revealed no significant difference between the female detection rate of SFOAE and that of the male detection rate at the *f*_p_ of 1 kHz and *L*_p_ of 50 dB SPL (χ^2^ = 0.003, *p* = 0.954, ES = 0.016). As shown in Table 1 and Figure 2c, SFOAEs were detected in 22 female participants (92%) and 17 male participants (65%). The female detection rate of SFOAE was obviously higher than the male detection rate at the *f*_p_ of 4 kHz and *L*_p_ of 30 dB SPL, and a significant difference existed between them (χ^2^ = 5.024, *p* = 0.025, ES = 0.668). As shown in Table 1 and Figure 2d, SFOAEs were detected in 15 female participants (63%) and 23 male participants (88%). At the *f*_p_ of 4 kHz and *L*_p_ of 50 dB SPL, the detection rate of SFOAE in males was obviously higher than that in females, and gender had a significant effect on the detection rate of SFOAE (χ^2^ = 4.612, *p* = 0.032, ES = 0.637).

### 3.2. Musical Training Influence

#### 3.2.1. PTA Test Results between Musicians and Non-Musicians

Figure 3 illustrates the mean values of PTA for musicians and non-musicians as a function of *f*_p_ value. Except for the PTA at an *f*_p_ value of 2 kHz, the mean hearing thresholds of musicians were lower than those of non-musicians at the other probe frequencies. A two-factor Scheirer–Ray–Hare test revealed that both group difference (musicians vs. non-musicians) and *f*_p_ had significant effects on hearing thresholds (group difference—*df* = 1, *H* = 5.194, *p* = 0.023; *f*_p_—*df* = 5, *H* = 15.243, *p* = 0.009), and interaction was found between the factors of group difference and frequency (*df* = 5, *H* = 11.841, *p* = 0.037). As shown in Table 2, simple effect analysis revealed that musicianship affected hearing threshold saliently at *f*_p_ values of 250 and 500 Hz (250 Hz—*df* = 1, *H* = 10.276, *p* = 0.001; 500 Hz—*df* = 1, *H* = 6.484, *p* = 0.011), and probe frequency affected hearing threshold saliently in the group of non-musicians (*df* = 5, *H* = 19.141, *p* = 0.002).

#### 3.2.2. SFOAE Amplitudes between Musicians and Non-Musicians

The mean values of SFOAE amplitudes for musicians and non-musicians as a function of *f*_p_ value at low [1 kHz; see Figure 4a] and high [4 kHz; see Figure 4b] probe frequencies are illustrated in Figure 4. As shown in Figure 4a, the mean SFOAE amplitudes of musicians were larger than those of non-musicians at all *f*_p_ values centered 1 kHz. The two-factor ANOVA tests revealed that musicianship had a significant effect on the SFOAE fine structure at the center frequency of 1 kHz (*df =* 1, *F =* 25.104, *p* = 0.000001, *ƞ*_p_^2^ = 0.075), while there was no significant difference among the SFOAE amplitudes of different *f*_p_ values (*df* = 10, *F* = 1.681, *p* = 0.084, *ƞ*_p_^2^ = 0.052). No interactions were found between the factors of musical training and frequency (*df* = 10, *F* = 0.334, *p* = 0.971, *ƞ*_p_^2^ = 0.011). As shown in Figure 4b, the values of SFOAE amplitudes between musicians and non-musicians at each *f*_p_ seemed to be slightly different, except for *f*_p_ values of 4040 Hz and 4160 Hz. The means and the standard deviations (means ± standard deviations) of SFOAE amplitudes of musicians and non-musicians were −7.942 ± 7.469 dB SPL and −9.492 ± 7.363 dB SPL, respectively. The two-factor Scheirer–Ray–Hare tests revealed that musicianship had no significant effect on the SFOAE fine structure at the center frequency of 4 kHz (*df* = 1, *H* = 3.107, *p* = 0.078). In addition, there is no significant difference among the SFOAE amplitudes of different *f*_p_ values (*df* = 10, *H* = 5.285, *p* = 0.871), and no interactions were found between the factors of musical training and frequency (*df* = 10, *H* = 1.547, *p* = 0.999).

### 3.3. Gain Function

Figure 5 illustrates the SFOAE input/output function (see Figure 5a) and the gain function (see Figure 5b) at 4 kHz for one subject. The gain function was transformed from the SFOAE input/output function by the SFOAE magnitude at each *L*_p_ subtracting (in dB) the corresponding probe level. The red line in Figure 5b represents the fitting curve of SFOAE strength. Three parameters, PS, CT, and CS, which were derived from this fit, are shown in Figure 5b. Both the input/output function and the gain function of SFOAE revealed the compressive characteristic of the cochlea. 

All participants’ three parameters and their linear regression fits at center frequencies of 1 and 4 kHz are shown in Figure 6. PS, CT, and CS are plotted as a function of probe frequency, which is displayed from the first to third panels. As shown in Figure 6a,b, the maximum SFOAE gain was similar between low and high frequency. For CT and CS, CT and CS grew larger at the frequency centered at 1 kHz, while they grew smaller at the frequency centered at 4 kHz as the probe frequency increased. The detailed statistical results are presented in Table 3. The independent-simple *t* tests revealed that probe frequency had a significant effect on CT (*t* = −3.861, *p* = 0.001, ES = 1.342) but no significant effect on PS (*t* = 0.120, *p* = 0.905, ES = 0.044). Additionally, the Mann–Whitney U tests revealed no significant effect on CS (*z* = −0.416, *p* = 0.677).

## 4. Discussion

In the present study, the PTA and the SFOAE amplitudes were tested in 65 normal hearing subjects (15 musicians and 50 non-musicians) at probe levels of 30 and 50 dB SPL in the center frequencies of 1 and 4 kHz to investigate the characteristics of SFOAEs, including detection rate, musical training influence, and gain function. The results indicated that (1) the detection rates of SFOAE were sensitive to gender, (2) musicians revealed enhanced hearing capacity and SFOAE amplitudes compared with non-musicians, and (3) probe frequency had a significant effect on the compression threshold of SFOAE. 

### 4.1. Detection Rate

In our study, the detection rates of SFOAE were tested for different genders at the conditions of different probe frequencies and probe levels. For gender, the detection rates of SFOAE between groups of females and males were compared under four conditions: 2 probe levels × 2 probe frequencies. Except for the *f*_p_ of 1 kHz and *L*_p_ of 50 dB SPL, gender had a significant effect on the detection rates of SFOAE. Under the condition of *f*_p_ = 4 kHz and *L*_p_ = 30 dB SPL, the female group’s detection rate of SFOAE was larger than that of the male group’s. The detection rate of SFOAE in males was larger than that in females under the conditions: (1) *f*_p_ = 1 kHz and *L*_p_ = 30 dB SPL, (2) *f*_p_ = 4 kHz and *L*_p_ = 50 dB SPL. The results in our study suggest that the detection rate of SFOAE is sensitive to the gender while the influence is irregular as the *f*_p_ and *L*_p_ change. These results are inconsistent with observations of SOAEs that the detection rate for female subjects is significantly higher than that for male subjects [52]. The effect of gender on the detection of SFOAE depends on probe frequency and probe level. This may due to the fact that SFOAE is a reflection emission generated by a single pure tone at a specific probe frequency. For low frequency, the detection rates of both females and males increased as the probe level grew. The reason for low detection rate of SFOAE at low probe level was possibly due to SFOAE, which overlapped the evoking stimulus in both time and frequency and was easily disturbed by background noise. For high frequency, the detection rate of males increased, but that of females decreased with probe level growing. From Figure 4, we can see that the SFOAE amplitude at 4 kHz (−9.492 ± 1.047 dB SPL) was lower than that at 1 kHz (−4.353 ± 1.649 dB SPL) (*t* = 8.724, *p* = 0.00000003, *ES* = 3.562). Thus, the detection rate of SFOAE may have become lower and more sensitive at high probe frequency. 

### 4.2. Musical Training Influence

In our study, we measured and analyzed the PTA and the SFOAE amplitudes between musicians and non-musicians to ascertain whether musical training affects the inner ear. Different statistical methods were used according to whether the data satisfied the normal distribution. Our results indicated that musicians exhibited lower hearing thresholds compared with non-musicians at 250 and 500 Hz. Several previous studies reported that musical experience improves perceptual and physiological frequency selectivity at the level of the cochlea [4,50]. PTA is a type of active listening that determines the individual’s minimum threshold of audibility. Our findings revealed that musical experience can also improve musicians’ audiometric thresholds as well as frequency resolution. For SFOAE, we observed that musicians exhibited larger amplitude when *f*_p_ = 1 kHz, while there was no significant difference for PTA between musicians and non-musicians at a probe frequency of 1 kHz. This suggested that it could be improved by musical training at the cochlear level. The factor that may account for these phenomena is the regulation of prestin, which is a motor protein responsible for outer-hair-cell electromotility [53]. As PTA contains the influence of behavior of the participant, and the SFOAE is an objective measure evoked within cochlea, it suggests that the SFOAE reflects a real cochlear difference compared with PTA. For SFOAE amplitude at high probe frequency, musical training had no significant effect. This may be related to hearing loss often occurring at high frequency because of noise and aging for both non-musicians and musicians [54]. 

### 4.3. Gain Function

Our results indicated that probe frequency had a significant effect on the compression threshold of SFOAE, revealing that the stimulus level at which SFOAE strength began to decrease was significantly higher at high frequency. It may have accounted for the response of the basilar membrane to sounds of different frequencies being strongly affected by its mechanical properties; at the base (high frequency), it was relatively narrow and stiff, while at the apex (low frequency), it was wider and much less stiff. Therefore, at high frequency, the nonlinearity of SFOAE appeared at higher probe level. 

As described in a previous study, SFOAE and DPOAE differ significantly in their compression features [44]. Owing to there being a different relationship of SFOAE and DPOAE metrics between normal and impaired hearing ears, the metrics combining SFOAE and DPOAE gain function are promising means to indicate hearing loss.

## 5. Conclusions

In summary, in the present study, the PTA and the SFOAE amplitudes were tested to investigate the characteristics of SFOAEs, including detection rate, musical training influence, and gain function. Our results provide a basis for laboratory investigation and clinical use of SFOAEs and demonstrate that musical training is beneficial to the rehabilitation enhancement of the auditory periphery and hearing threshold.

## Figures and Tables

**Figure 1 brainsci-09-00255-f001:**
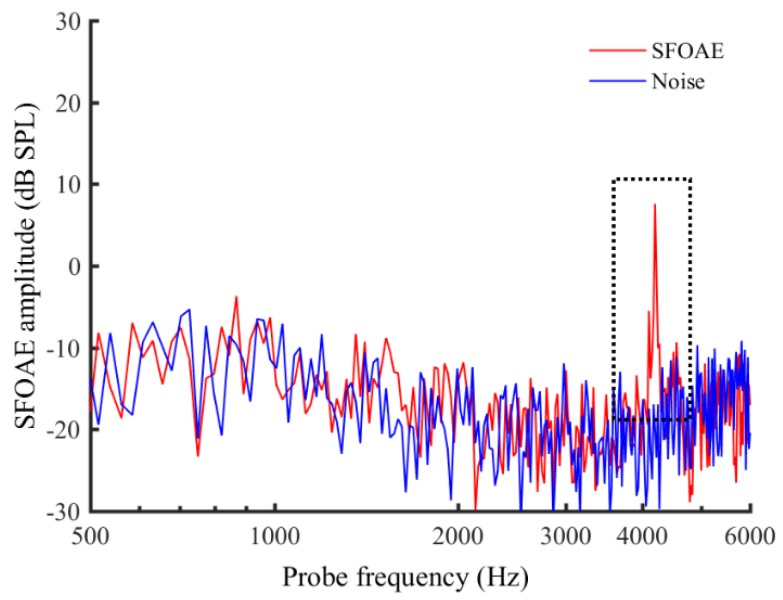
An example of the presence of stimulus frequency otoacoustic emission (SFOAE). Probe level was 50 dB sound pressure levels (SPL), and the center frequency (CF) is 4 kHz. Red line indicates SFOAE and blue line indicates the background noise. The presence of SFOAE is represented in the black dashed box.

**Figure 2 brainsci-09-00255-f002:**
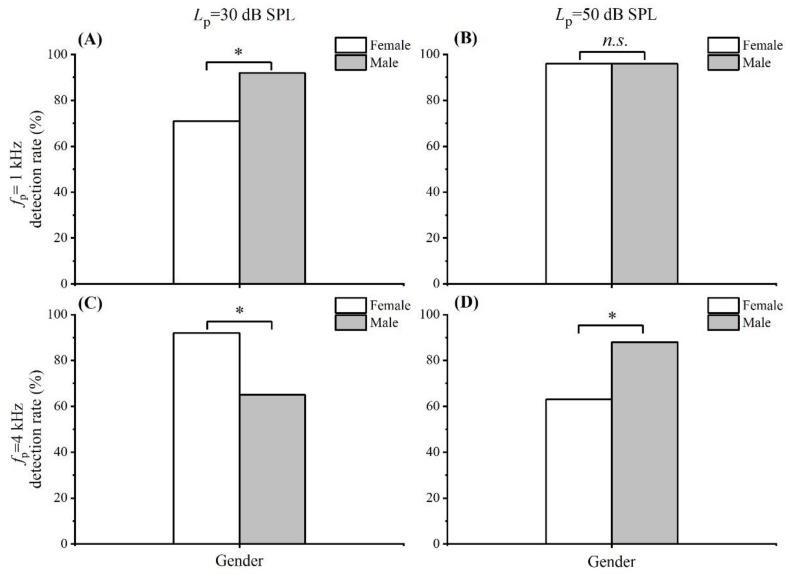
The detection rates of SFOAE between groups of female and male under four conditions: 2 probe levels × 2 probe frequencies. (**A**) *f*_p_ = 1 kHz; *L*_p_ = 30 dB SPL; (**B**) *f*_p_ = 1 kHz; *L*_p_ = 50 dB SPL; (**C**) *f*_p_ = 4 kHz; *L*_p_ = 30 dB SPL; (**D**) *f*_p_ = 4 kHz; *L*_p_ = 50 dB SPL. * *p* < 0.05, *n.s*., not significant (chi-square tests).

**Figure 3 brainsci-09-00255-f003:**
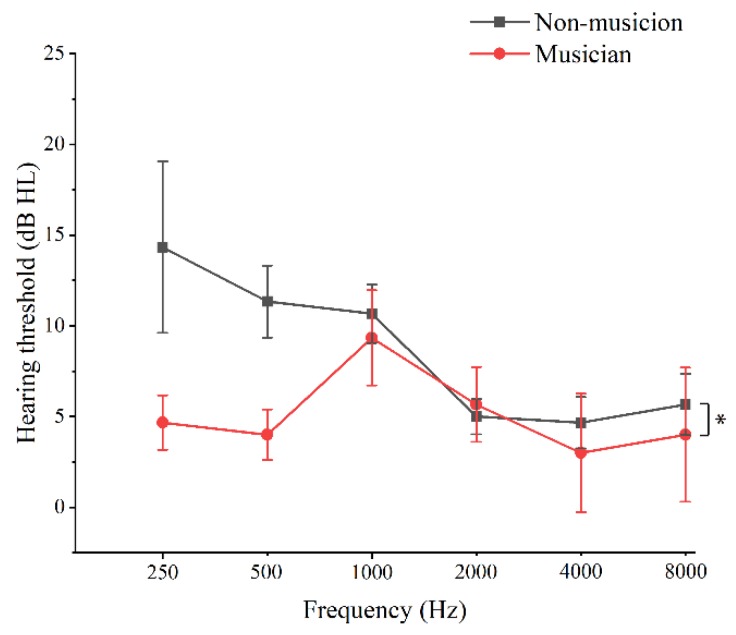
The mean values of pure tone audiometry (PTA) for musicians and non-musicians as a function of *f*_p_s. * *p* < 0.05, *n.s*., not significant (two-factor Scheirer–Ray–Hare test). Vertical bars represent the standard errors of mean.

**Figure 4 brainsci-09-00255-f004:**
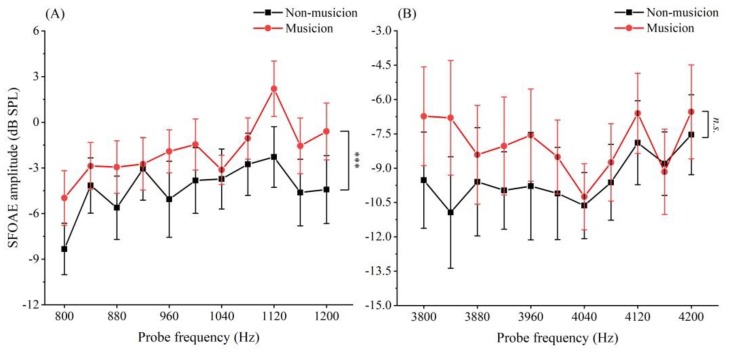
The mean values of SFOAE amplitudes for musicians and non-musicians as a function of fps. (**A**) Mean SFOAE fine structure at the probe frequency centered low frequency: 1 kHz; (**B**) mean SFOAE fine structure at the probe frequency centered high frequency: 4 kHz * *p* < 0.05, ** *p* < 0.01, n.s., not significant (two-factor Scheirer–Ray–Hare test). Vertical bars represent the standard errors of mean.

**Figure 5 brainsci-09-00255-f005:**
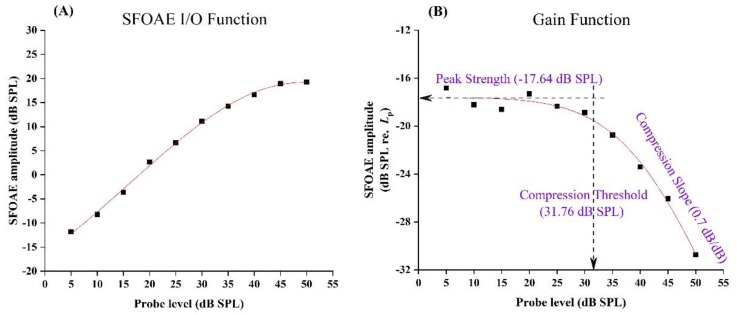
The SFOAE input/output function and the gain function at 4 kHz for one subject. (**A**) The SFOAE input/output function; (**B**) the SFOAE gain function. The gain function was transformed from the SFOAE input/output function by the SFOAE magnitude at each Lp subtracting (in dB) corresponding probe level. The red line represents the fitting curve of SFOAE strength. Three parameters—peak strength (PS), compression threshold (CT), and compression strength (CS)—were derived from this fit.

**Figure 6 brainsci-09-00255-f006:**
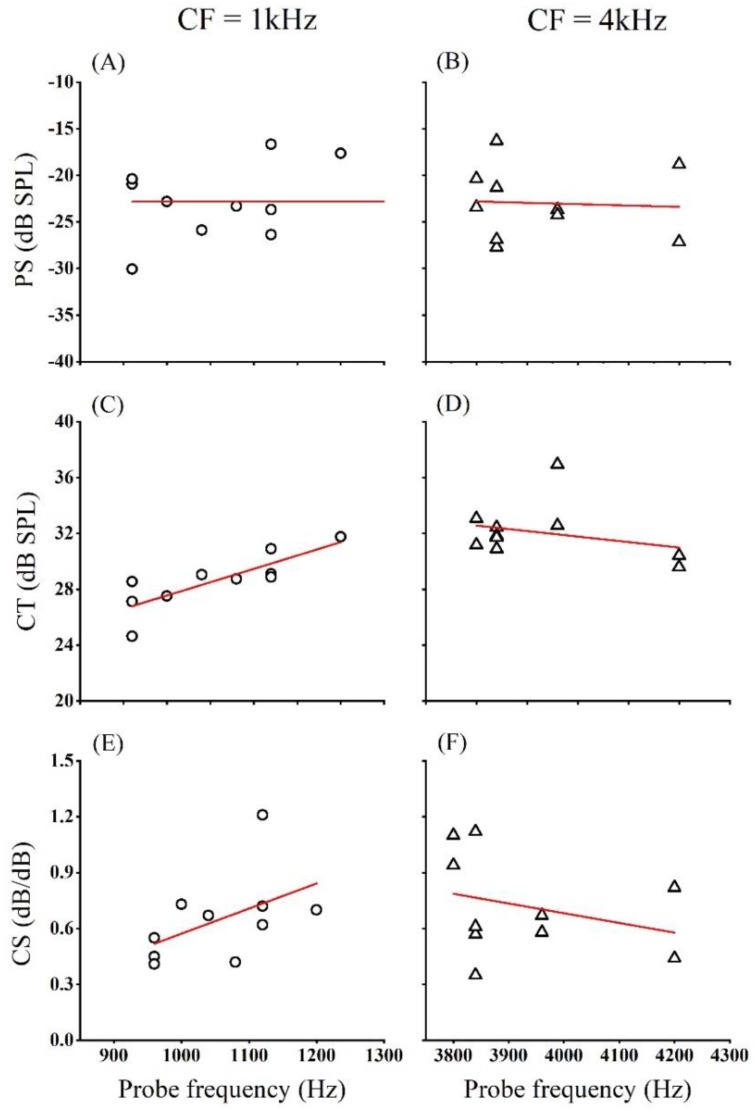
All participants’ three parameters and their linear regression fits at the CFs of 1 (left) and 4 kHz (right). PS, CT, and CS are plotted as functions of probe frequencies (from first to last panel). The triangles denote the values of PS, CT, and CS at the CFs of 1 kHz. The circles denote the values of PS, CT, and CS at the CFs of 4 kHz. (**A**) PS at 1 kHz; (**B**) PS at 4 kHz; (**C**) CT at 1 kHz; (**D**) CT at 4 kHz; (**E**) CS at 1 kHz; (**F**) CS at 4 kHz.

**Table 1 brainsci-09-00255-t001:** The detection rates of SFOAE with the different factors of gender at the conditions of probe frequency and probe level.

Factor	Condition	Group ^a^	Detection Ratio (%)	χ*^2^* Value	*P* Value	*ES*
Gender	*f*_p_ = 1 kHz *L*_p_ = 30 dB SPL	F	71	3.899	0.048 *	0.583
		M	92			
	*f*_p_ = 1 kHz *L*_p_ = 50 dB SPL	F	96	0.003	0.954	0.016
		M	96			
	*f*_p_ = 4 kHz *L*_p_ = 30 dB SPL	F	92	5.024	0.025 *	0.668
		M	65			
	*f*_p_ = 4 kHz *L*_p_ = 50 dB SPL	F	63	4.612	0.032 *	0.637
		M	88			

^a^ F denotes female, M denotes male; ES, effect size; * *p* < 0.05.

**Table 2 brainsci-09-00255-t002:** Simple effect results between the factors of group difference and frequency.

Frequency (Hz)	Group	df	*H* Value	*p* Value
250	/	1	10.276	0.001 **
500		1	6.484	0.011 *
1000		1	0.077	0.781
2000		1	0.179	0.672
4000		1	0.076	0.783
8000		1	0.012	0.913
/	musicians	5	7.943	0.159
	non-musicians	5	19.141	0.002 **

* *p* < 0.05, ** *p* < 0.01.

**Table 3 brainsci-09-00255-t003:** Statistics results of PS, CT, and CS at *f*_p_ values centered 1 and 4 kHz.

Metrics	Mean (SD) at 1 kHz	Mean (SD ^a^) at 4 kHz	*t*/*z* Value	*p* Value	ES
PS	−22.773(4.080)	−22.985(3.789)	0.120	0.905	0.054
CT	28.627(1.969)	32.062(2.010)	−3.861	0.001 **	1.754
CS	0.648(0.233)	0.720(0.266)	−0.416	0.677	/

^a^ SD denotes standard deviation. ** *p* < 0.01.

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
