# Peer review of "Characteristic of Stimulus Frequency Otoacoustic Emissions: Detection Rate, Musical Training Influence, and Gain Function"

_brainsci, 2019, doi:10.3390/brainsci9100255_

Round 1

Reviewer 1 Report

see attachment

Reviewer 2 Report

Overall This manuscript describes 3 Experiments that explore the interaction of certain factors on audiological outcomes. 1) Examining gender, probe frequency, and probe level on SFOAE detection rates; 2) examining the impact of musicianship on pure tone thresholds and SFOAE amplitudes; 3) examining the role of probe frequency on various SFOAE gain-function descriptors. Some areas can be improved, but mostly the Discussion and Introduction need re-writing. The introduction provides an adequate background, but it does not justify the Experiments. While the Discussion has many claims that are not supported by the Results. These details are listed below. Needs editing by a native English speaker, as much of the writing is unclear. Introduction Overall, the Introduction reads like a series of statements, without a clear flow of ideas that establishes the reason for carrying out these 3 particular experiments. Only in the 2nd to last paragraph is the motivation touched upon. Please re-write the introduction to be a persuasive argument that establishes a need for your experiments. 1, ln. 30: Definitions of SOAE and EOAE are backwards 2, ln. 42: OAE is not a “method” 2, ln. 42: OAE is better [at indicating cochlear function] that what? Pg 2, ln 50: this sentence does not make sense Pg 3, ln 81: Please provide a reference for the claim “[The gain function] reveals the OAE generation mechanism” mechanism. Pg 3, ln 87: center frequency should say 1-4 kHz (not ¼ kHz = 250 Hz) Results Figure 1: Since A and B are N.S., in the text, you can NOT describe one factor as “higher” than the other. They are the same, within measured natural variation. Pg 6, ln 206: There is no description in the methods regarding the measurement of the SFOAE fine structure. This must be added if it is included as an outcome. Figure 4: Can you label the parts of the gain function to match the acronyms PS, CT, and CS? (probably change “compression strength” to “compression threshold”) Discussion Ln 264: your results from experiment 1 do NOT indicate that female detection rates of SFOAE are higher than those of males – in fact, you found that there was NO SIGNIFICANT difference Ln 275: not “especially” at 250 and 500 hz, but ONLY at 250 and 500 hz and not elsewhere Interestingly, these are the frequency regions where phase encoding is dominant, which might bear mentioning Ln 278: you claim that your results identified “variations of intensity”, but that is not an outcome of any of the Experiments Ln 279-80: this claim is unsubstantiated Ln 285-6: this claim is most likely incorrect, and not supported by evidence or citation Ln 289-301: remove

Reviewer 3 Report

Lines 106-108: This is probably fine as is, but the implied confidence intervals for musical training commencement and musical training duration would seem to imply the possibility of a subject starting musical training after age 10 or undergoing musical training for less than 15 years (especially since the mean for the latter is less than 15). One possibility could be to publish the maxima and minima for both measures for both musician and nonmusician subjects. One could also create a graph with those distributions for the supplemental materials. My last suggestion would be to use more nuanced phrasing for describing those distributions that does not invite the potential for ambiguity.

Lines 147-152: If this is beyond the scope of this paper by all means change nothing, but I was curious as to the possible reasons why PTA and SFOAE amplitudes at 4kHz aren't approximately normal while they are at 1kHz. A couple sentences in the discussion about that could be helpful for those not intimately aware of the various tests used in this paper, but it is probably not necessary.

Lines 15, 87, 256: since "1/4 kHz" can be interpreted as one quarter kilohertz or just 250 Hz, I'd consider revising it to "1 and 4 kHz" or "1 or 4 kHz" or something to that effect

Line 82: " In this study..." could perhaps be the start of a new paragraph.

Lines 187 and 202: The italics for the two sub-titles "PTA" and "SFOAE amplitudes" aren't terribly clear, especially for the former. Assuming it's consistent with journal-specific style guidelines, I'd suggest using colons instead of periods as well as making "PTA" into "PTA amplitudes" or something like that to make it clearer that it's a title of some sort and not an accidental fragment of the text proper.

Line 235: As far as I can tell, the only place that Peak Strength, Compression Strength, and Compression slope are written out in their non-abbreviated form is in Figure 4B and at the bottom of page 4/top of page 5. I would suggest writing them out in the caption as "Peak Strength (PS)" and so forth.

Along similar lines, in Line 247 I would add "(CF)" after "central frequency"

Lines 307-308: The phrase starting at "joint-gain function metrics..." sounds awkward. Perhaps some revision like "...metrics are promising means to indicate..." would be appropriate.

Round 2

Reviewer 1 Report

Characteristic of stimulus frequency otoacoustic emissions: detection rate, musical training influence, and gain function

General

The paper has been improved. However, one of the concerns was the poor link between the three studies described in the paper. In particular the Discussion still fails to discuss the outcomes with respect to each other. Further, as mentioned below, the Results on detection rate should be reduced in Figures and text.

Major comments

1) The detection rate data are all described in Figure 2. Figures 3 and 4 are not needed, and should be removed. The analyses should be performed on the data as presented in Figure 2. There seem to be interactions between gender, frequency and level. The description in the Results section should be limited: lines 224 – 243 may be deleted.

In the Discussion the authors only repeat the Results without really discussing the observations. For instance, why would SFOAEs if detected at low levels not be detected at high levels (as apparantly for several female participants was the case)?

Minor comments

Introduction

Line 63. ‘previous literatures’ to be replaced by ‘previous studies’

Methods

Line 149. Delete ‘SPL’. dB SPL indicates an absolute sound level defined as dB relative to 20 µPa. Here the level of the signal is described relative to the level of the noise, thus in dB.

Results

Line 204. Delete ‘< 0.05’ since p=0.048 is sufficient. See also other places, delete ‘p<0.05’ and ‘> 0.05’ if p value is mentioned.

Reviewer 2 Report

These revisions addressed each of my concerns. 

Minor:

I still believe that this sentence is awkward with the inclusion of the phrase "especially only". If phase encoding is going to be mentioned, it might be interesting to the reader to expound on that slightly, or just remove it. 

“Our results indicate that musicians exhibit smaller lower hearing thresholds compared with non-musicians, especially only at 250 and 500 Hz, and these are the frequency regions where phase encoding is dominant.”
